ecology/e-science

Citizen science, open science, vignette survey

**Author for correspondence:**
Muki Haklay
e-mail: m.haklay@ucl.ac.uk

# Contours of citizen science: a vignette study

Muki Haklay[1], Dilek Fraisl[2], Bastian Greshake Tzovaras[3], Susanne Hecker[4,5,6], Margaret Gold[7], Gerid Hager[2], Luigi Ceccaroni[8], Barbara Kieslinger[9], Uta Wehn[10], Sasha Woods[8], Christian Nold[1], Bálint Balázs[11], Marzia Mazzonetto[7], Simone Ruefenacht[7], Lea A. Shanley[12], Katherin Wagenknecht[13], Alice Motion[14], Andrea Sforzi[15], Dorte Riemenschneider[7], Daniel Dorler[16], Florian Heigl[16], Teresa Schaefer[9], Ariel Lindner[3], Maike Weißpflug[6], Monika Mačiulienė[17] and Katrin Vohland[18]

[1]Department of Geography, University College London, Gower Street, London WC1E 6BT, UK
[2]International Institute for Applied Systems Analysis (IIASA), Schlossplatz 1, Laxenburg 2361, Austria
[3]Université de Paris, INSERM U1284, Center for Research and Interdisciplinarity (CRI), 8bis Rue Charles V, Paris 75004, France
[4]Helmholtz Centre for Environmental Research - UFZ, Department of Ecosystem Services, Permoserstr. 15, 04318 Leipzig, Germany
[5]German Centre for Integrative Biodiversity Research (iDiv) Halle-Jena-Leipzig, Puschstr. 4, 04103 Leipzig, Germany
[6]Museum für Naturkunde—Leibniz Institute for Evolution and Biodiversity Science, Berlin, Germany
[7]European Citizen Science Association (ECSA), c/o Museum für Naturkunde, Invalidenstr. 43, Berlin 10115, Germany
[8]Earthwatch, Mayfield House, 256 Banbury Road, Oxford OX2 7DE, UK
[9]Centre for Social Innovation GmbH, Linke Wienzeile 246, Vienna 1150, Austria
[10]IHE Delft Institute for Water Education, Westvest 7, AX Delft 2611, The Netherlands
[11]Environmental Social Science Research Group (ESSRG), Ferenciek Tere 2, Budapest 1053, Hungary
[12]Nelson Institute, University of Wisconsin-Madison, 550 N Park ST, Madison, WI 53706, USA
[13]Technische Hochschule Wildau, Hochschulring 1, Wildau 15745, Germany
[14]The University of Sydney, Australia
[15]Maremma Natural History Museum, Strada Corsini 5, Grosseto 58100, Italy
[16]University of Natural Resources and Life Sciences, University of Vienna, Gregor Mendel Strasse 33, Vienna 1180, Austria
[17]Kaunas University of Technology, K. Donelaičio g. 73, Kaunas LT-44249, Lithuania
[18]Naturhistorisches Museum Wien, Burgring 7, Vienna 1010, Austria

MH, 0000-0001-6117-3026; BGT, 0000-0002-9925-9623; LC, 0000-0002-3116-0811; CN, 0000-0002-5922-8291; BB, 0000-0001-6937-499X; MaMa, 0000-0003-4109-6371; FH, 0000-0002-0083-4908; MW, 0000-0002-6082-5333

Citizen science has expanded rapidly over the past decades. Yet, defining citizen science and its boundaries remained a challenge, and this is reflected in the literature—for example

in the proliferation of typologies and definitions. There is a need for identifying areas of agreement and disagreement within the citizen science practitioners community on what should be considered as citizen science activity. This paper describes the development and results of a survey that examined this issue, through the use of vignettes—short case descriptions that describe an activity, while asking the respondents to rate the activity on a scale from 'not citizen science' (0%) to 'citizen science' (100%). The survey included 50 vignettes, of which five were developed as clear cases of not-citizen science activities, five as widely accepted citizen science activities and the others addressing 10 factors and 61 sub-factors that can lead to controversy about an activity. The survey has attracted 333 respondents, who provided over 5100 ratings. The analysis demonstrates the plurality of understanding of what citizen science is and calls for an open understanding of what activities are included in the field.

# 1. Background: introduction

Over the past decade, the field of citizen science has rapidly expanded [1]. From early identifications by Bonney [2] and Irwin [3], the field has grown; piquing the interest of policy makers, research funding organizations, scientists and those wishing to harness the field for knowledge generation and public engagement with science. Concurrently, citizen science has evolved from its historical roots, leading to debate around which projects, activities or initiatives constitute citizen science, which exist at the nexus with other forms of research, and which are perhaps mislabelled by their initiators or participants.

A number of calls and attempts have been made to create a modern definition of citizen science [4]. Consensus on a common definition is, however, difficult to reach for an interdisciplinary endeavour that is so broad and placed at the intersection of numerous scientific fields. Furthermore, there may be a need for a discipline- or context-specific definition (such as for a specific funding call). Instead of seeking consensus, therefore, we sought to encompass the plurality of views from citizen science practitioners and affiliated communities and to identify the common characteristics that practitioners expect from a citizen science activity. The views of people are from citizen science practitioners, science communicators or policy officers in the area of research and innovation with an interest in open science matter. This multidisciplinary target group is currently contributing to the emerging discussion on what activities are included in the field or not.

This can also help the development of context-specific definitions. To identify these common characteristics, we opted to use the methodology afforded by a vignette study, where miniature case studies were presented to survey respondents (most of them citizen science and public engagement practitioners) who rated each according to their personal views. This study not only showed a wide divergence in views and opinions on whether an activity does or does not constitute citizen science but also revealed specific areas of agreement and disagreement about aspects of citizen science, for example, the role of commercial companies in the practice of citizen science or the level of cognitive engagement that is expected from participants.

Furthermore, each miniature case study, or vignette, provided an opportunity to contextualize activities and to gather citizen science characteristics that were based on authentic case studies inspired by real-world initiatives, as opposed to abstract assessment of principles. The divergence and plurality of views reflected in these vignettes form the foundation of the European Citizen Science Association (ECSA) Characteristics of Citizen Science [5]. They are intended as a stimulus for discussion and debate, a tool to identify the characteristics in citizen science projects and a useful framework that builds upon the ECSA 10 Principles of Citizen Science [6].

## 1.1. What is citizen science?

The current use of the term 'citizen science' sprang from two different epistemological viewpoints based on their field of origin. The first conception of citizen science originated with Alan Irwin [3,7] and focuses on the role of citizens as stakeholders of the outcomes of research, such as in the public and environmental factors of health. Irwin situates citizen science 'at the point where public participation and knowledge production—or societal context and epistemology—meet, even if that intersection can take many different forms' [8]. Such approaches, Irwin argues, provide an opportunity to bring members of the public and science closer to consider the possibilities for a more active 'scientific citizenship', with an explicit link to public policy. The second conceptualization of citizen science by

Rick Bonney [2] focuses on volunteers and their contributions to field-based observations of the natural world, facilitated through the coordination efforts of professional scientists. Bonney highlights that in a research field such as ornithology, the role of volunteers who participate in data collection is integral to how the research operates. His work has contributed to the growth of citizen science, especially in the USA and within environmental projects.

More recently, the term citizen science has been commonly used to describe different forms of participation in scientific knowledge production. In this sense, it overlaps with a wide array of terms that are used to describe various forms of participatory action research and digital volunteerism, including Community Science, Civic Science, People-Powered Science, Participatory Mapping, Participatory Science, Volunteered Geographic Information (VGI), Community Remote Sensing, Citizen Observatories, Crisis Mapping and Citizen Generated Data, the last gaining acceptance by distinct organizations of the United Nations and amongst statistics and data communities. The variety of scientific and monitoring activities in which the public can participate, the range of disciplines and the diverse organizational and cultural contexts in which they are deployed have contributed to the evolution of these varying terms. In each case, authors have sought to grasp the phenomenon in their respective contexts; revealing as much about their own interests and area of focus as about the unique features and different types of citizen science practices. Additionally, it is important for project leaders or initiators to communicate with participants in their choice of terms, as these should be able to facilitate a shared understanding of the aims and ethos of the activity, and the context of the participants [9].

A scientometric meta-analysis performed by Kullenberg and Kasperowski in 2016 [1] discovered a large number of terms that could fall under the citizen science umbrella and clustered them around three main focal points: (i) collecting and analysing biology, conservation, biodiversity and climate change data, (ii) collecting geographical data and (iii) public participation in social sciences and epidemiological research. Based on patterns of scientific publications, the fields of natural science, social science and geography emerged as the best-represented categories, with a particularly high scientific output in astronomical journals [1]. A similar and more recent bibliometric analysis conducted by Bautista-Puig *et al.* [10] retrieved 5100 publications on citizen science and showed a cumulative average yearly growth rate of just over 16%, significantly higher than the Web of Science database average growth rate of about 5% in those same years. Their keyword co-occurrence-based clustering identified four main subject groups with a range of frequently occurring terms: (i) health: 'participatory action research', 'community-based research' and 'action research', (ii) biology: 'biodiversity monitoring'; (iii) geography: 'volunteered geographical information', 'participatory GIS' and 'public participation GIS' and (iv) public: 'public participation' [10].

The US National Science Foundation (NSF) (e.g. [11] suggested 'Public Participation in Scientific Research (PPSR)' as a more inclusive term instead of 'citizen science', owing to the perception in the US context that 'citizen' science is limited to legal citizenship and not inclusive of others (e.g. immigrants, tourists, etc.). There was also concern about the perceived association of the term citizen science solely with informal education. Nevertheless, the term PPSR has not gained wide acceptance. Thus, while it is important to acknowledge the sensitivities around the term citizen science, there is a risk of introducing unused labels when seeking a new, all-encompassing term.

### 1.1.1. Typologies of citizen science in the literature

The diversity of terms, conceptualizations and definitions has also led to a proliferation of typologies of citizen science. Each typology represents a different viewpoint on citizen science and uses different aspects to describe it. In table 1, we present a range of typologies that have been proposed in the literature, with a description of their orientation and the classifications that were chosen to describe the differences and ranges in citizen science practice. These typologies and the classifications they offer are not uniformly structured, and they can be applied in different ways. When related to a citizen science case, they can be used as a specific descriptor for the entire activity (e.g. a project can be regarded as either consulting, contributory, collaborative, co-created or collegial according to [14]), or they can help describe different stages or possible outcomes of a citizen science activity.

As outlined in table 1, many of the authors take a matrix approach, which is useful for adding nuance and insight to describing the complexity of citizen science, as well as highlighting the evolution of typology development itself. It also reinforces the need and desire to employ a multi-faceted view when characterizing, describing and analysing citizen science practice more generally.

**Table 1.** Different typologies of citizen science and citizen science-related activities.

| Terminology used | Orientation and focus | Classifications within typology |
|---|---|---|
| Citizen Science | **Multi-dimensional** focus on types of the activity | Action-oriented, conservation, investigation, virtual and education [12] |
| | **Knowledge producer** and **activity/project goal and focus** | Matrix approach: Citizens or researchers as main knowledge producers, addressing a research question or intervention in a socio-ecological system [13] based on [12,14] |
| | **Nature of the participatory task** | Passive sensing, volunteer computing, volunteer thinking, environmental and ecological observations, participatory sensing and civic/community science [15,16] |
| | **Learning dimensions** | Learning of project mechanics, pattern recognition skills, on-topic extra learning, scientific literacy, off-topic knowledge and skills and personal development [17] |
| | **Complexity of the citizen science approach** and **participation structure** | Matrix approach: Elaborate approach vs. simple approach, and mass participation vs. systematic monitoring, and in addition computer-based projects [18] |
| | **Communication goals** of a citizen science project | Goals of communication messages from citizen science projects: Awareness, Conversion, Recruitment, Engagement, Retention [19]) |
| | **Education aspects** | Increasing interest in science, using scientific tools, specific disciplinary content, scientific reasoning, to developing an identity in science and more [20] |
| | **Multi-dimensional focus** on the nodes of engagement | Behavioural activities, affective/feeling, learning/cognition and social/project connections [21] |
| | **Activity type and epistemic practice** | Sensing, computing, analyzing, self-reporting, making [22] |
| Public Participation in Scientific Research (PPSR) | **Relational aspects and role definitions**, with implicit information on **depth of involvement** | Consulting, contributory, collaborative, co-created, and collegial [14] which is an expansion of [11] |
| Citizen Engagement in Social Innovation | **Direction/goal of a project and scale** (based on number of participants) | Matrix typology: Investigating present states to developing future solutions; from few to many participants [23] |
| Citizen Science and Volunteered Geographic Information | **Engagement of participants** in an activity | Crowdsourcing, distributed intelligence, participatory science and extreme citizen science [24] |
| Citizen science and environmental management | **Relationship type** and **type of activity encounter** | Matrix approach: Cooperative vs. adversarial relationships and deliberate vs. serendipitous [25] |

*(Continued.)*

| Terminology used | Orientation and focus | Classifications within typology |
|---|---|---|
| Citizen Science and Conservation | **Type of projects/ formats of citizen science** | Bioblitzes, ongoing monitoring programmes, bounded field research and inventory projects, data processing projects [26] |
| Citizen Observatories | **Multi-dimensional** for a systematic review framework | Geographic scope, type of participants, establishment mechanism, revenue stream, communication paradigm, effort required, support offered, data accessibility, availability and quality [27] |
| Citizen Science and Innovation Management | **Business model of the project and its funding** | Motivated individual; Small Crowdsourcing; Outreach; Research and Innovation (R&I); and Long Term NGO [28] |
| Citizen Science in Health and Biomedical Research | **Research focus** and **modes of participation** | Observational and Interventional research; matrix approach to participation models: Professional driven vs. Public driven and Independent participation vs. Collective participation, resulting in Traditional science, N-of-1/DIY science, N-of-many-1's/contributory and N-of-we/co-created participation modes [29] |
| Community Based Monitoring | **Multi-dimensional** on the aspects that can influence the establishment and functioning of a CBM | Goals and objectives of the project, technologies, participation, power dynamics [30] |
| Policy and Citizen Science | **Policy outcomes and impact** | Policy outcomes—from addressing a local environmental nuisance, to monitoring national policy and the stages of the policy cycle: issue identification, measure identification, implementation, monitoring (effectiveness) [31] |

### 1.1.2. Descriptive aspects of citizen science

In addition to the typologies and discussions in the literature, we have further identified the following aspects that can be relevant when describing a specific citizen science initiative:

*Types of participants:* Citizen science activities can engage a wide range of participants and members of the public. These can include, among others, school pupils, visitors of museums, adventurers or ecotourists, activists and amateur experts in their free time.

*Scientific fields:* Citizen science takes place and can be rooted in many different scientific fields and areas of research, such as life sciences, physical sciences, medical research, engineering, social sciences and humanities. Considerations of outputs and outcomes, citizen science outcomes can range from knowledge outcomes, such as journal articles, or information used by participants to address issues of local concern, to practical policy outcomes and tangible outputs can range from an open data repository to a personal checklist of nature observations.

*Open science dimensions:* The growing importance of open science [32] and the integration of citizen science as part of the European Open Science conceptualization [33] for opening up scientific processes stresses the need to address and consider open science practices in citizen science, such as the use of open data, open access publication, releasing research code as open source and open lab/workflows.

*Technology use and accessibility:* The type of technology and its use (e.g. pen/paper, desktop computer, mobile phones, sensor boxes, scientific instrumentation (binoculars, telescopes, DNA sequencing kit)) as

well as their accessibility in terms of cost of acquisition or access and the skill levels required to operate them, are important aspects to consider (cf. Gharesifard *et al.* [30]).

*The temporal dimension*: Citizen science activities and projects can range from an activity that happens only once (one-off), over a short-term (a few days or weeks), infrequently (once a month or less) and/or long-term (every day and/or over a long period of time) (cf. Ballard *et al.* [26]).

### 1.1.3. What is not citizen science?

While some exclusionary statements appear within the typologies that we have reviewed there is no explicit definition of what should not be regarded as citizen science. Nevertheless, we provide examples of exclusions that are noted in the literature. These exclusions are helpful in identifying areas of controversy or ambiguity and are useful in the overall aims of this research. We have grouped them into specific areas and present recurring concepts and arguments in the literature that are decisive for the assessment of an activity being citizen science, or not: activeness, engagement, the profile of the participant, knowledge production and data transparency.

One of the frequent aspects to emerge in the discussion is the **level of activeness** within a project, with several publications asserting that volunteers must have an active role in elements of the research process for a project to be considered citizen science. For example, for Wiggins & Crowston [12] such an active role does not include the provision of computing resources (sometime called Volunteer Computing) and Heigl *et al.* [4] exclude data contribution in the form of 'opinion polls or data collection on participants' (p. 8091). Strasser *et al.* [22], on the other hand, include volunteer computing as a presumably passive form of participation, and Haklay *et al.* [16] include 'passive sensing', which allows for participation through automatic data capture. Similar activeness is linked to the nature of contribution, for example, by excluding projects that collect data that were already shared on social media platforms [34].

A second source of differentiation is the need for identifying the role of **learning and engagement** within the project. Pocock *et al.* emphasize that 'engagement alone is not citizen science. Perhaps you have an important message to convey but with no need to gather data. There are many examples of engagement working really well to raise awareness of a particular issue by communicating with many people without it being citizen science' [35]. Similarly, learning without engagement is not considered citizen science [36]. In their analysis of extension programmes (US government-operated programmes that provide educational learning experiences to farmers), Ryan *et al.* [37] point out that 'in the context of agriculture, the missions of the Extension are to bring science and technology to farmers and food producers and to learn about new observations and problems from those stakeholders. *This bidirectional flow of knowledge itself is not citizen science*, but it creates an opportunity to do citizen science—generates new knowledge, through partnerships' (p. 2, emphasis added).

A third area of contention is the **meaning of volunteering**. The US Federal Community of Practice on Crowdsourcing and Citizen Science (FedCCS), US EPA, NASA, UNEP, UNESCO and the German and the UK Parliament Offices for Science & Technology emphasize volunteering as an integral part of citizen science within their definitions [38], meaning participation based on free choice and without monetary compensation. However, the US National Academy of Sciences, Engineering, and Medicine (NASEM) [20] highlights that there is also semi-voluntary participation in citizen science as part of education curricula (either at school or at university). In addition, Resnik *et al.* [39] as well as Fraisl *et al.* [40] mention other forms of compensation and payment, such as reimbursement of expenses, as well as direct payment to participants.

Next, we can see the importance of the **formal training of participants**. For example, the NASEM report that was noted above [20] asserts that a project focusing on water quality 'where only professional water quality technicians collect the data would not [be considered citizen science]. A project where students collect water quality data solely for their own edification does not fit the committee's description of citizen science.' There is also an issue of intention: 'a project where people play a video game (however much that game is dealing with real scientific problems like protein folding) is not citizen science unless the players know they are dealing with real scientific challenges, have some understanding of those challenges and the relevant science and know that their individual results are useful' (pp. 1–2). This illustrative debate gives an idea of how restrictive the description of citizen science can be in certain applications. Activities that exclusively involve people with domain-specific professional and scientific backgrounds should be considered 'not citizen science'. If their involvement is outside of what they've professionally been trained for (e.g. collecting data in ancillary/related domains), arguably they could still be citizen scientists.

**Power relations** are also important to consider. Del Savio *et al.* [41], in their analysis of uBiome and A/BGP, point out that such projects should be considered 'not citizen science' if they do not democratize science. More specifically, they state that this is because participants have very little involvement in the design or management of the project. The authors conclude that 'importantly, scientists and entrepreneurs opting for crowdsourcing will not assess the success of their projects on the basis of the quality of citizen engagement that they are able to promote. *Citizen-science projects are often designed by actors motivated by very different hopes than to democratize science. Hence we should be cautious when assessing the participatory rhetoric of citizen-science promoters …*' (p. 13, emphasis added). This point is also emphasized by Chen [42].

According to some authors, for a project to be classified as citizen science, **data transparency** is needed, where 'transparency' is understood according to Wiggins and Wilbanks's discussion of aggregated self-quantifying data or so-called N-of-1 studies [29]. The authors proffer that 'N-of-many-1's data collection projects skirt close to being considered "not citizen science", particularly when recruitment and enrolment of participants are conducted in ways that do not clearly disclose details around data access and participant benefits prior to registration' (p. 9). Nevertheless, there are many instances where citizen science data are not open access due to privacy, safety, sensitivity or sovereignty concerns; see, for example, [43].

In summary, the outlined typologies of citizen science from the literature, the additionally identified descriptive aspects and arguments for or against something being citizen science, help to understand the diversity and variability within the field. Such variability also indicates that there may be disagreement on what is considered citizen science, when different aspects are taken together. These insights have helped us to develop the set of descriptive factors of citizen science that are presented in §2. They have served as an initial baseline for discussion.

# 2. Survey rationale and methodology

To identify the range of views and opinions about citizen science activities, a survey that is based on miniature case studies (vignettes) was deemed the most appropriate. Vignette studies have been used widely in healthcare and social studies (see [44,45]) and allow the elicitation of perceptions, expectations, opinions, impressions or values around complex situations, based on the presentation of hypothetical situations.

As demonstrated earlier, a wide range of typologies and classifications of and debates around citizen science exist that provide a basis for the identification of different dimensions of citizen science activities. Based on these, a set of descriptive factors that can influence the decision about the classification of an activity as citizen science, or not, were identified (table 2). Apart from factors 1 and 6, which are ordinal, the rest of the factors are categorical with some element of order in their sub-factors. These factors are as follows:

   (i) Activeness—the level of cognitive engagement.
  (ii) Compensation—the financial relationships between the activity owner and the participant, addressing the issues of volunteering and crowdfunding.
 (iii) Purpose of the activity—the overall aim of the activity.
 (iv) Purpose of knowledge production—the aim and application of the knowledge that was produced in the activity.
  (v) Professionalism—the skills requirements from the participants.
 (vi) Training—the level of training provided to participations.
(vii) Data sharing—the conditions under which the resulting data is shared.
(viii) Leadership—the type of organization or individual who leads the activity.
 (ix) Scientific field.
  (x) Involvement—the degree of participation in different stages of a hypothetical process on the basis of [14,46].

Each factor was divided into sub-factors. Sub-factors that were expected to be disputed or controversial are highlighted (e.g. since some of the literature excludes volunteer computing from citizen science on the basis of the passive cognitive engagement [12], sub-factor 1.3 is highlighted). In total, there are 61 sub-factors, of which about half (30) can be considered controversial.

There are several options to find out the community views regarding these factors. For example, it is possible to directly survey these 61 sub-factors and ask the extent to which each of them influences a decision about identifying an activity as citizen science, or not. However, when viewed and assessed

**Table 2.** Descriptive factors for vignette development, those with higher controversy potential highlighted in bold.

| Factor | Categories and explanation |
| --- | --- |
| 1 Activeness | 1.1 Active—requires full cognitive engagement during participation |
| | **1.2 Semi-active**—limited cognitive engagement (e.g. responding to short alerts in a micro-task) |
| | **1.3 Passive**—no engagement beyond set-up |
| 2 Compensation | 2.1 Volunteer—unpaid participation |
| | 2.2 Expenses—only expenses are paid |
| | **2.3 Small incentives**—minimal payment or partial payment which is indirect to the activity (e.g. for coordinating, providing equipment for community-based monitoring that can be used for other purposes) |
| | **2.4 Payment for the activity** |
| | **2.5 Crowdworking**—small payment for tasks |
| | **2.6 Subscription fee**—participants pay to participate in a activity |
| | **2.7 Student**—compulsory part of studies |
| 3 Purpose of the activity | 3.1 Scientific/research—scientific or research focused activity |
| | 3.2 Policy outcome—e.g. environmental management monitoring, action or other policy actions |
| | **3.3 Public engagement**—the main purpose is engagement |
| | **3.4 Education**—focus on education outcomes |
| | **3.5 Game**—focus on gaming environment |
| | **3.6 Reuse of social media**—reuse of images or other information that was submitted in social media |
| 4 Purpose of knowledge production | 4.1 Scientific discovery—producing a scientific paper |
| | 4.2 Scientific management—producing data for policy |
| | 4.3 Personal discovery—personal level learning |
| | **4.4 Local knowledge sharing**—sharing local lay knowledge within the community (not necessarily with researchers) |
| | **4.5 Alternative knowledge**—non-science knowledge: e.g. perceptions and opinions |
| | **4.6 Commercial knowledge**—for commercial applications |
| 5 Professionalism | 5.1 Anyone—no assumption about expertise |
| | 5.2 Self-selected—a barrier to entry or assumptions about prior knowledge |
| | **5.3 Targeted**—aiming at a specific set of experts, for activities beyond their work |
| 6 Training | 6.1 No training/light training—the activity is open to anyone and does not require training beyond immediate participation |
| | 6.2 Significant training—the activity requires prior training and possibly accreditation as a condition for participation |
| | **6.3 Academically focused**—the activity requires participants to have a higher education degree |
| | **6.4 High skills**—the activity expects participants will hold higher degrees (MSc/PhD) to participate |
| | **6.5 Specialists**—the activity is aimed at specialists |

| Factor | Categories and explanation |
| --- | --- |
| 7 Data sharing | 7.1 Open scientific/research data—collected by scientists/research institute and shared openly |
| | 7.2 Scientific data—collected by scientists/research institute but not shared |
| | **7.3 Education/engagement only**—undertaken as part of education/engagement activity and outputs not used beyond this activity |
| | **7.4 Commercially aggregated (N-of-many-1s)**—data that is collected by commercial actors, such as health and activity data |
| | **7.5 Collected by non-professional(s), not shared** |
| | **7.6 Public Authorities data**—in monitoring activities, where data is delivered to authorities (shared or not shared) |
| | **7.7 Integration with official data** |
| | **7.8 Data aggregation**—integration of data from multiple activities |
| | **7.9 Voluntary personal data**—sharing personal data with researchers (e.g. health research, consumer behaviour research, mobility research) |
| 8 Leadership | 8.1 Scientists/researchers—led by scientists or researchers or a research institution |
| | **8.2 Individual**—self-led by an individual, with herself as the only participant. |
| | 8.3 Community—community-led |
| | **8.4 Commercial**—led by a commercial company |
| | 8.5 Public sector—led by people who work in the public sector (e.g. environmental officers) |
| | 8.6 CSO—led by a civil-society organization such as a non-governmental organization (e.g. environmental charity) |
| 9 Scientific field | 9.1 Life and Medical Science |
| | 9.2 Earth Science |
| | 9.3 Formal Science |
| | 9.4 Natural Science |
| | 9.5 Social Science |
| | **9.6 Humanities** |
| | **9.7 The Arts** |
| | 9.8 Inter/Trans/Multidisciplinary |
| 10 Involvement | 10.1 Multiple stages |
| | **10.2 Single stage—issue/topic identification/research question setting** |
| | **10.3 Single stage—research design** |
| | 10.4 Single stage—research tool/methods development |
| | 10.5 Single stage—data gathering |
| | 10.6 Single stage—data analysis and interpretation |
| | **10.7 Single stage—data sharing and/or results communication** |
| | **10.8 Single stage—policy design/management decision** |

without a context, it will be impossible to understand the nuance and the interplay of factors—and the purpose of the study, as indicated, was to identify and illuminate the spectrum of views. Another potential approach is to identify cluster of factors as they appear within existing examples from the literature and only ask participants to survey them. While this is an improvement on the study of the factors alone, such an approach is likely to lead to oversimplification of existing situations. We therefore aimed at identifying a survey approach that will allow the respondents to evaluate the activity within a context and to focus on cases that are likely to raise a discussion. We therefore concluded that an approach in which a short case description is presented to the respondents, and

they need to indicate the degree to which it can be classified as a citizen science activity is the most appropriate. As a result of these considerations, we developed the survey as a vignette study.

All the factors play a role in citizen science activities, and—as highlighted in the previous section—some of the discussions in the literature show concern about the relationship between different factors [12]. For example, the appropriateness of a payment to participants is context-dependent [47]. Therefore, we aimed to understand how the factors operate in context. To do so, we presented short case studies inspired by real-world activities to the survey respondents and asked for their view regarding the degree to which they would classify this as a citizen science activity. The use of vignettes provided us with an opportunity to present examples of activities that were representative of the complexities of citizen science without the need to explicitly state which sub-factor or combination of sub-factors is explored. This ensured that we received answers that took into account the full context and richness of citizen science practice and not the abstract classification of specific aspects of activities.

To arrive at a reasonable number and variety of vignettes, we developed a strategy to guide the construction and selection of case descriptions based on the factors described earlier. The primary goal of the research was to identify and better understand the controversial, or 'grey', areas within citizen science activities using both fractional factorial analysis (see [44]) and qualitative text analysis. Hence, we focused on cases that would represent the different controversial sub-factors to varying degrees and constructed vignettes that ensured the inclusion, and subsequently the testing, of these aspects. In addition, we included some vignettes that represented widely agreed citizen science activities and some that would probably not be considered citizen science.

Of the 50 vignettes, we created 10% of the vignettes as clear examples of citizen science based on the literature, 10% to be illustrations of activities that are not citizen science (e.g. clinical trials or surveys, as noted in §1.1.3), and 80% of cases mixed a combination of controversial and non-controversial elements from the different factors. We also created vignettes based on published examples of citizen science activities, and on examples we considered to be on the verge of being considered citizen science, to ensure that the vignettes were representative of real cases where possible. Table 3 showcases a sampling of the vignettes (along with the factor vectors and controversial sub-factors in bold). The complete set of vignettes used in the study can be found in the supplementary material. Case descriptions were based on website information (e.g. from the British Trust of Ornithology or Wikipedia), the experience of team members (e.g. from the GROW Observatory project) or other published material. To provide consistency across cases and to allow respondents to be reasonably able to look through several examples and to classify them, we kept the length of each vignette to 70–100 words. In addition to a description of the activity, each vignette also contained the following fictitious elements: a participant's name, location, some background of the participant in the activity and details about the activity owner. Each vignette was peer-reviewed by at least one member of the research team.

The vignettes in table 3 represent the range used for the study. V3 (for Vignette 3), V12, V13 and V15 are examples for vignettes that address a specific area of controversy—from financial contribution to an activity as the sole involvement, to participation in an activity that produces commercial knowledge. V41 and V45 are vignettes that represent widely accepted citizen science activities (V41 describes Galaxy Zoo, which is widely celebrated [51]), while V47 is a clinical trial activity that is frequently described as 'not citizen science' (§1.1.3).

Respondents had to rate, using a scale bar from 0 to 100, to what degree they would identify each vignette as citizen science. Since the study aimed to establish the collective view on each vignette (and therefore the sub-factors), rather than the individual views of each respondent, it was acceptable for different people to respond to a different set of vignettes. The survey was therefore set to display the vignettes in a random order and allow respondents to choose to complete the survey at any point or progress to the next vignette. This approach also ensured that the effort of respondents is distributed across vignettes for those who did not assess all cases but rated more than one vignette. We estimated that reading and considering a vignette would take approximately 1 min. Assuming respondents would dedicate 10–15 min for a survey, we anticipated that each response would include about 10 vignettes. A pilot run of the survey showed that respondents had varying levels of confidence when assessing the cases. We considered this useful to capture in the evaluation of their rating and, therefore, added three levels of confidence to the survey (easy, somewhat complex and difficult).

The survey response form also offered two optional text fields: (i) to provide a name for the activity, so different terminologies within citizen science could be identified and (ii) to justify the rating that was given to the vignette. At the beginning of the survey, very few details were requested from the respondent: a description of their role (research, public sector, private sector, policy, private citizen, NGOs, other), years of experience in citizen science (none, up to 1 year, 1–5 years, 5–10 years and

**Table 3.** Examples of vignettes.

| | Case description | Sub-factors | Source |
|---|---|---|---|
| 3 | Jane is a long-time supporter of the charity British Trust of Ornithology (BTO) work, as she cares about birds. She is an active supporter of the Garden Birdwatch programme (GBW) and happy to give it £17 a year. However, she doesn't have time to carry out the birdwatching survey. She is reading with interest the reports from the BTO GBW and finds the information motivating to continue her support of the project. | 1.1, **2.6**, 3.1, 4.2, 5.1, 6.2, 7.2, 8.6, 9.1, 10.7 (subscription fee) | Inspired by [48] |
| 12 | Jacques has joined a massive multiplayer game for which he pays a subscription fee. In the game, he is travelling through galaxies trading, mining resources and competing with other players. He enters an area, where he classifies human proteins, for which he gets credits that can be traded throughout the game. The project was initiated by scientists and a gaming company. The classifications will eventually get published in the human protein atlas. | 1.1, 2.6, **3.5**, 4.1, 5.2, 6.1, 7.1, 8.1/**8.4**,9.1, 10.6 (game) | Inspired by [49] |
| 13 | Dorota is a photographer in Katowice, Poland, and she specializes in sharing images of interesting wildflowers and insects on Flickr, where information about the location and time is recorded with the image. She is taking part with groups of photographers who are interested in the beauty of insect photography. Lena, an ecologist at the university, is scanning these groups regularly and using the images to identify invasive species—some of which are captured because they are often unfamiliar or visually interesting. Lena uses her findings with public authorities to support environmental management and also comments on Flickr to communicate with the photographers. | 1.1, 2.1, **3.6**, 4.2, 5.1, 6.1, **7.6**, 8.1, 9.1, 10.5 (reuse of social media) | Inspired by [50] |
| 15 | Erik is a teacher in Uppsala, Sweden. For the past 15 years, he is running a weather station that is part of the Weather Underground's Personal Weather Station Network with over 250 000 participants who share their observation data, just like Erik. In return for the data sharing, the company is providing tech support, data management services and customized, free-of-charge access to forecasts. The company uses the data to produce a global weather forecast as a commercial service. | 1.1, 2.1, 3.2, **4.6**, 5.1, 6.1, **7.4**, **8.4**, 9.2, 10.5 (commercial knowledge) | Inspired by [16] |

*(Continued.)*

**Table 3.** *(Continued.)*

| | Case description | Sub-factors | Source |
|---|---|---|---|
| 41 | Femke is a teaching assistant in Eindhoven, the Netherlands. She has heard about a website where you can help astronomers by classifying images of galaxies. She didn't expect to get hooked on the experience, but after a few classifications, she finds that looking at these images is fascinating and in doing so, she has learnt new things about the universe and the composition of galaxies. She is dedicating significant time every evening to classify galaxies on the website. The results of her analysis will be used by the scientists who developed the platform to publish important scientific papers. | 1.1, 2.1, 3.1, 4.1, 5.1, 6.1, 7.2, 8.1, 9.4 (clear citizen science—Galaxy Zoo) | Inspired by Hanny von Erkel—see [51] |
| 45 | Sebastian lives in Hanover and is a hobby gardener with a local allotment garden. Last year, he did an online course on regenerative growing and signed up to a European wide growing experiment, comparing a polyculture with a monoculture setup. He followed instructions given to him and set up the experiment on his plot. He joined online meet-ups with other experimenters and collected data from his site and shared it via an online form. He also analysed his data himself and shared it via social media. He received the accumulated results of all experiments and joined a final online discussion. He also agreed to be named as a contributor in an academic paper about the experiment. | 1.1, 2.1, 3.1, 4.1/4.3, 5.2, 6.2, 7.1, 8.1, 9.4 (Clear citizen science) | Inspired by [52] |
| 47 | Yanis is a bus driver in Greece. He suffers from arthritis, a chronic condition and was offered to participate, voluntarily, in a study about a new physiotherapy technique to manage his condition. He is asked to use an app to report on his symptoms several times a day. The study is run by medical researchers at his local hospital, and the results will be published in an open-access journal article. | Not citizen science—Clinical trial | Created for this study |

**Would you call this a citizen science activity?**

Sandra from Birmingham, England, recently had her first child. On a forum that is dedicated to issues of using detergents with cloth nappies, she found a group of other young parents on Facebook, and together they are carrying out a double-blind test of different detergents and their impact on nappies. The results of the study are shared widely through a medical charity and can influence the National Health Service recommendations for using these nappies.

\* How confident are you about judging this case?

○ It is easy to decide     ○ It is somewhat complex to decide     ○ I find it difficult to decide

\* To what degree would you identify this as citizen science?

0%                          50%                         100%

How would you call this activity? **(optional)**

Why did you give that rating? **(optional)**

**Figure 1.** An example of the survey layout.

more than 10 years), and a field to describe their disciplinary background. At the end of the survey, questions about the country in which the respondent works and an option to be acknowledged and cited by name in the study were provided. In terms of design, it was expected that the responses would present a pattern of participation inequality [53], and therefore, a large number of respondents would evaluate a single vignette, a medium number would respond to 10–15 and a very small group would rate all cases. We therefore assumed that from 100 respondents, approximately 60 would rate a single vignette, 30 would rate ten vignettes and ten respondents would rate 50 vignettes, resulting in an average of 17 ratings per vignette, allowing for an indication of high or low agreement. We also hypothesized that the majority of vignettes would be categorized as either 'citizen science' or 'not citizen science', with only a minority showing ambiguity, following the literature that was reviewed above, which represents clear categories and delineations. It is noteworthy that this assumption about the pattern of response means that it will not be applicable to compare the responses of the participants but to look at the rating that each vignette received. Since the vignettes are displayed to the respondents in a random order, the majority of them—and especially those that will only classify a few vignettes—will see a different set from other people who categorized very few cases. As we will see in the analysis, we do not segment the participants' response beyond their experience in the field.

Following an evaluation of a range of survey tools, SurveyMonkey was chosen because it provided a slider response interface, the ability to order the vignettes randomly, and for respondents to stop at any point (figure 1). The survey was launched on 11 December 2019, and was closed on 26 December 2019. The promotion of the survey was carried out through social media (Twitter, Facebook, LinkedIn) and via mailing lists in the fields of citizen science, science communication, ecology and general scientific interest. As noted earlier, the recruitment targeted people who are already within areas that are using and developing citizen science. Despite the growth in citizen science that we have described earlier, there are fields with deeper familiarity with it. To make an informed decision if an activity should or should not be part of citizen science, some knowledge of citizen science is necessary. However, attempts were made to reach different disciplines and areas where citizen science is active, so as to ensure a wide range of perspectives from those that are using and developing citizen science.

Once the survey was completed, the data were prepared for analysis—including the removal of duplicate submissions (see §3 for details) and the removal of all identifying details. The two option

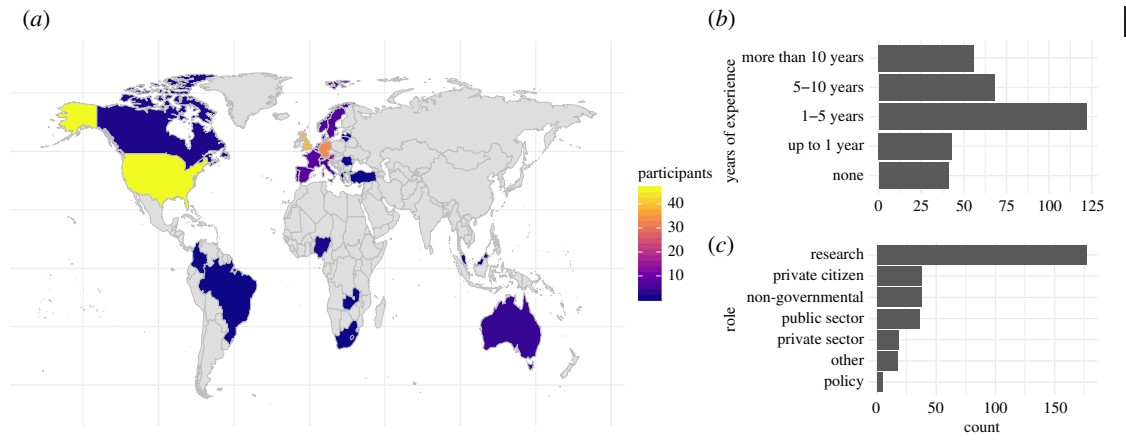

**Figure 2.** Characteristics of the survey respondents: (*a*) distribution of respondents across countries in which they currently live/work, (*b*) years of experience respondents have with citizen science and (*c*) role in which they are taking the survey.

fields provided a significant amount of textual information, which was separated to support qualitative analysis (see §5).

# 3. Survey respondent characteristics

We recorded 392 respondents to our survey, 59 of whom did not include any vignette answers and were therefore excluded from further analysis. A total of 13 duplicates were identified, of which nine were in the previously excluded set, and the remaining four were found to contain substantially different responses and were therefore counted in the total final dataset of 333 respondents.

A geographical location was provided by 213 respondents, most of whom indicated the USA or Western Europe, but there were also responses from Eastern Europe, South America, Africa and Asia (figure 2*a*). The respondents furthermore covered a broad spectrum of prior experience in citizen science, with 25% of them having none or less than a year of experience with citizen science and 33% having 1–5 years of experience (figure 2*b*). The majority of respondents (53%) described their role as being in academic research. Participation from private citizens, people working in the public sector, or non-governmental organizations made up for around 11% of respondents each (figure 2*c*).

The distribution of how many vignettes the respondents rated is broad and shows a skew towards the two extremes—40 respondents only rated a single vignette and 42 rated all 50 vignettes. The median number of rated vignettes was 11 (mean: 15.48) and 37 respondents rated at least 20 vignettes (figure 3).

As the order of the vignettes was randomly assigned for each survey respondent, we achieved a uniform number of responses across all 50 vignettes. Overall, the number of responses per vignette ranged from 90–115, with no systematic differences between the experience levels or roles of the respondents for each of the vignettes (figure 4).

# 4. Views about citizen science

Collectively, the respondents gave 5155 ratings across the 50 different vignettes, with all possible degrees of citizen science represented in the ratings. A similar skew towards the extremes is shown in the distribution of ratings, with 23.5% of the answers rating the degree of citizen science at 100 and 16% of answers giving a degree of zero (figure 5*a*). Furthermore, we see a clear outlier for the rating of 50—which is given in 6.3% of all answers—indicating respondents' level of indecision.

Vignettes which respondents found easy to rate were classified as 'definitely not citizen science' (0) or 'definitely citizen science' (100), while vignettes that were found to be somewhat complex or difficult to decide tended towards ratings of 50 (figure 5*b*). Overall, 68% of all answers were judged to be easy decisions, while less than 5% were considered difficult (figure 5*c*).

The perceived degree of citizen science varied drastically between the different vignettes (figure 5*d*). Some vignettes—such as V45, V43 and V42—were consistently judged as 'citizen science' by the survey participants, while other vignettes—e.g. V3, V49, V27—were overwhelmingly given ratings of zero,

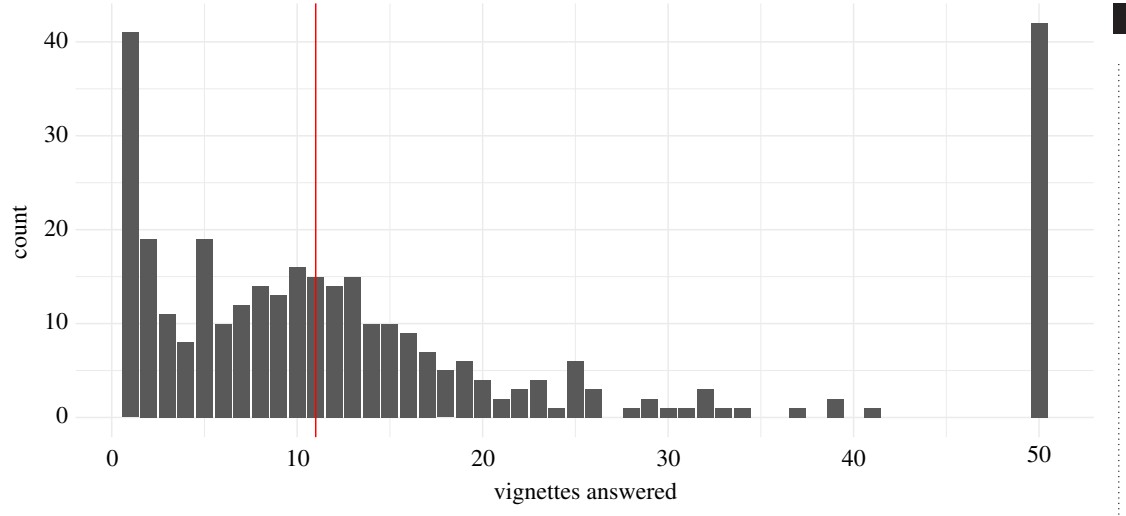

**Figure 3.** Number of vignettes rated by respondents. Red line indicates the median.

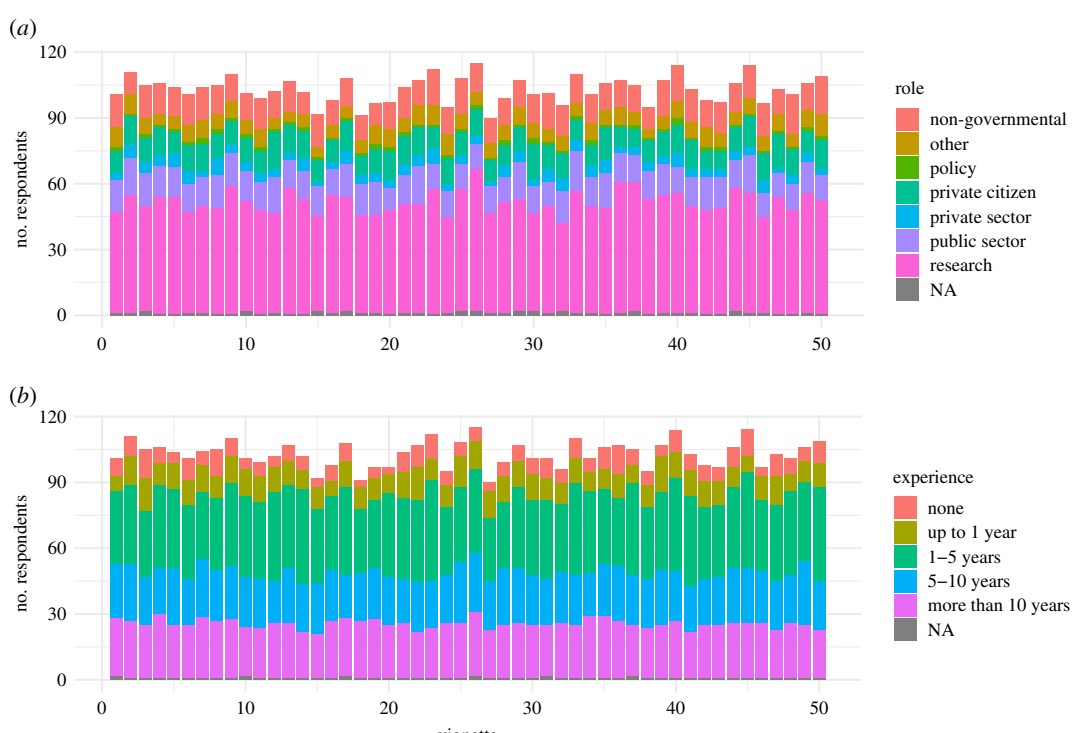

**Figure 4.** How often each vignette was rated and the (*a*) role and (*b*) experience level of the respondents that rated each vignette.

indicating that they are seen as 'not citizen science'. Furthermore, we observed a number of vignettes— e.g. V12, V13 and V47—in which the answers given by participants were spread across the whole range of possible degrees of citizen science.

We investigated the ambiguous cases further, to first test whether prior differences in respondents' experience with citizen science resulted in significant rating differences, by broadly categorizing the respondents into two groups—those with 0–1 years of experience and those with 1–10 years of experience—and comparing the ratings between these groups for each vignette using a Mann–Whitney–Wilcoxon test and a Dunn–Bonferroni correction for multiple testing. The extent of prior citizen science experience only had an impact on a few individual vignettes with contrasting ratings (figure 6*a*).

Following this we investigated the impact of how easy or hard people found it to judge the degree of citizen science, dividing the answers into 'easy' and '(somewhat) complex' or 'difficult'. Using the

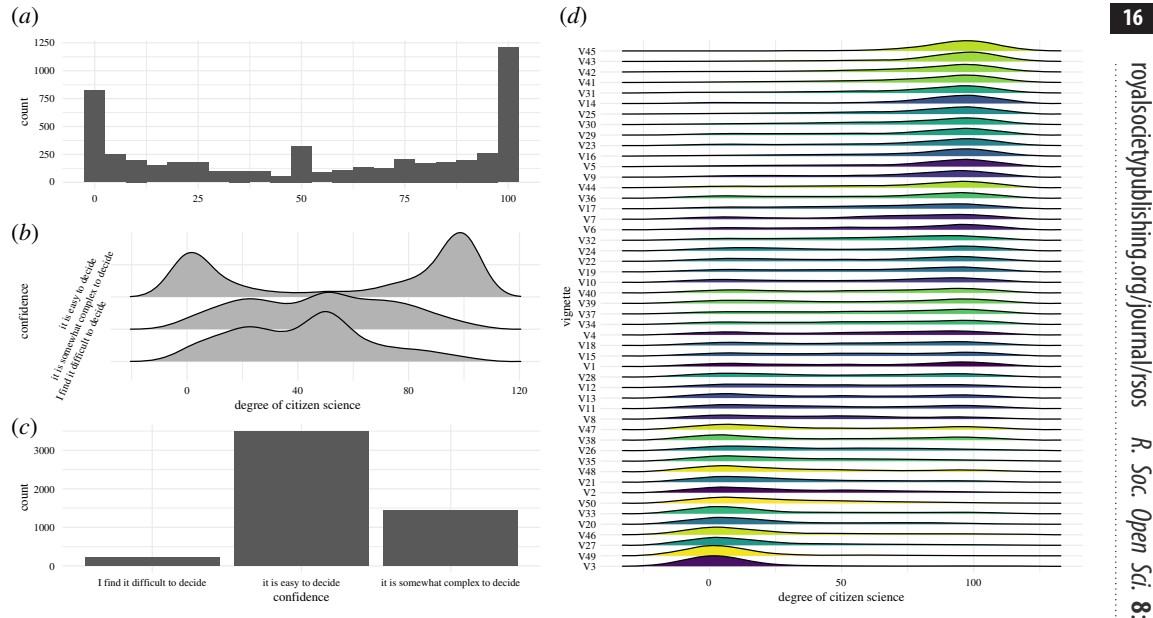

**Figure 5.** Overview of the ratings provided by the respondents: (a) degree of citizen science given by people over all 50 vignettes, (b) ratings depending on the level of confidence, (c) respondents' confidence ratings and (d) degree of citizen science ratings per vignette—vignettes are ordered by their median rating.

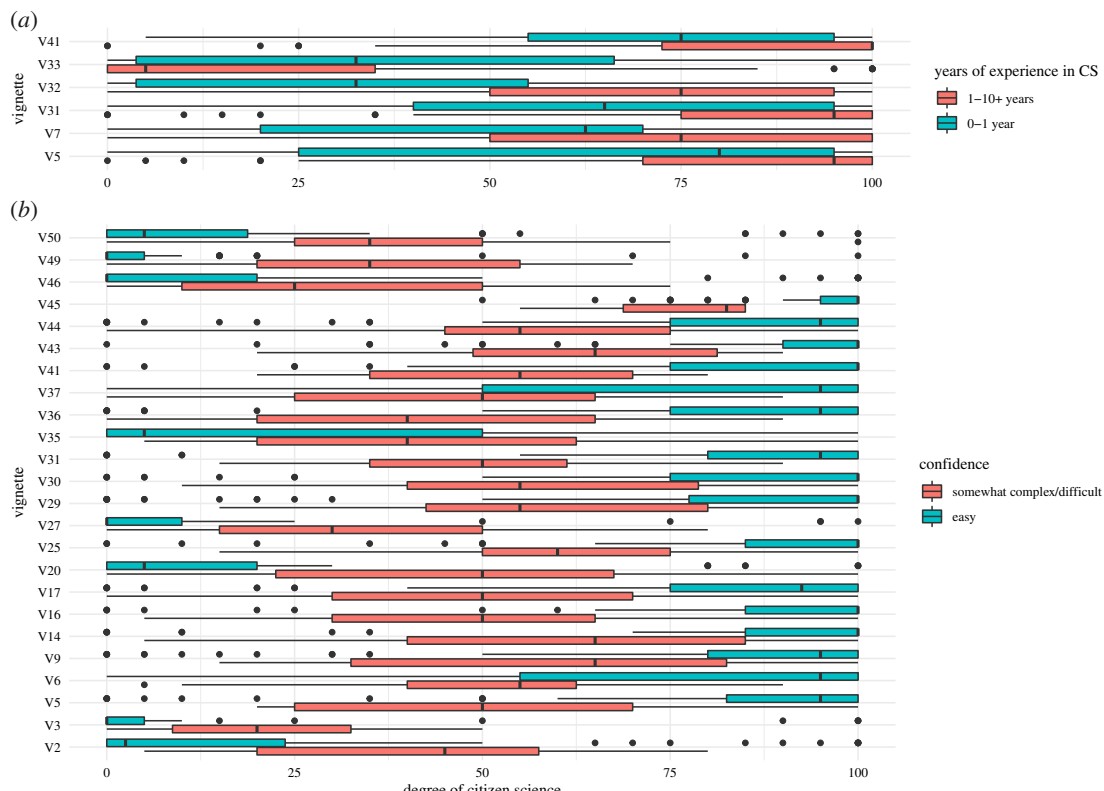

**Figure 6.** Vignettes with tendencies for significance and significant differences in ratings based on (a) respondents' experience and (b) confidence.

Mann–Whitney–Wilcoxon test, we found that 24 vignettes (figure 6b) were rated at significantly different degrees of citizen science between these two confidence groups (post-Dunn–Bonferroni correction $p \leq 0.05$). In all of these cases, we observed that the ratings of less confident respondents tended towards a neutral rating, which is in line with what we observed in the overall distributions (figure 5b).

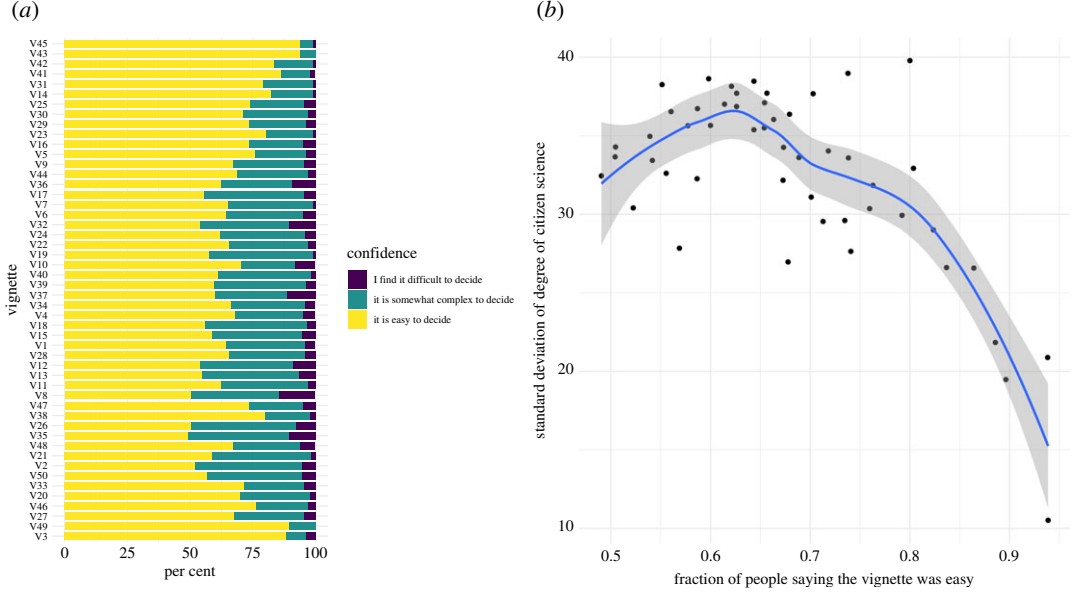

**Figure 7.** Respondents' (*a*) confidence and (*b*) its correlation with the ratings of the different vignettes.

While 70% of the respondents with more than 1 year of prior citizen science experience reported finding it easy to judge the vignettes, 60% of respondents with less than 1 year reported the same.

The degree of confidence also varied between the different vignettes, ranging from 50 to 94% of people saying the vignettes were easy to rate (figure 7*a*). We found a significant relationship between the percentage of people agreeing that a vignette was easy to rate and the distribution of the degree of citizen science for that vignette, where the deviation in ratings significantly decreased with increasing confidence of the respondents. Furthermore, below a certain confidence value (40%) the deviation in ratings started to decrease, as more respondents defaulted to a neutral rating (figure 7*b*).

# 5. Content analysis of textual responses

## 5.1. Selection of study sample and analysis design

The textual comments provided to the vignettes in the survey served as data for the qualitative analysis. We analysed text material from seven vignettes to illustrate and gain better understanding of the range and nuance of opinions and views about the different cases described in the vignettes. They were selected based on having a similar total number of ratings for each vignette; altogether, the seven vignettes had 722 ratings with 245 textual comments (see table 4). We provide here the qualitative analysis based on a selected set of text responses. The full set of text responses to the vignettes can be found in the supplementary material.

After a first overview of the textual information provided in free text to each of the chosen vignettes, three coders conducted a three-step mixed-method qualitative content analysis after [54]. First, we analysed respondents' comments independently and inductively to identify argument patterns that were grouped into thematic categories. In a second step, we complemented the categories with a deductive analysis of their accordance with the distinct factors relevant to the respective vignette (table 4). Third, we compared the independent analysis and synthesis results through joint discussions until we reached an agreement of interpretation.

## 5.2. Content analysis

*Compensation—subscription fee* (*Vignette 3*) received the most 0 ratings, i.e. 'not citizen science', and was consistently considered easy to rate. The main argument for low ratings is repeated in various re-formulations that financial support is not considered an active involvement in the scientific process. Various comments from respondents stressed that public financial support for and consuming

**Table 4.** Selected vignettes for qualitative content analysis with similar numbers in 0 and 100 ratings.

| vig. no. | total ratings | no. textual comments | avg. other ratings | % 0 rating | % 100 rating | % other ratings | factors (controversial factors are highlighted in bold and factor titles are provided) |
|---|---|---|---|---|---|---|---|
| 3 | 105 | 38 | 18.1 | 63.81 | 2.86 | 33.33 | 1.1, **2.6**, 3.1, 4.2, 5.1, 6.2, 7.2, 8.6, 9.1, 10.7 <br> **2.6 Subscription fee** |
| 12 | 102 | 41 | 45 | 11.76 | 15.69 | 72.55 | 1.1, 2.6, **3.5**, 4.1, 5.2, 6.1, 7.1, 8.1/**8.4**, 9.1, 10.6 <br> **3.5 Game—focus on gaming environment, 8.4 Commercial—activity led by a commercial company** |
| 13 | 107 | 48 | 45.2 | 19.63 | 15.89 | 64.49 | 1.1, 2.1, **3.6**, 4.2, 5.1, 6.1, **7.6**, 8.1, 9.1, 10.5 <br> **3.6 Reuse of social media—reuse of images or other information that was submitted in social media, 7.6 Public Authorities data—in monitoring activities, where data is delivered to authorities (shared or not shared)** |
| 15 | 92 | 37 | 45.7 | 9.78 | 23.91 | 66.3 | 1.1, 2.1, 3.2, **4.6**, 5.1, 6.1, **7.4**, **8.4**, 9.2, 10.5 <br> **4.6 Commercial knowledge, 7.4 Commercially aggregated, 8.4 Commercial** |
| 41 | 103 | 23 | 66 | 1.94 | 45.63 | 52.43 | 1.1, 2.1, 3.1, 4.1, 5.1, 7.2, 8.1, 9.4 Clear citizen science—Galaxy Zoo |
| 45 | 114 | 26 | 85.2 | 0 | 57.02 | 42.98 | 1.1, 2.1, 3.1, 4.1/4.3, 5.2, 6.2, 7.1, 8.1, 9.4 Non-controversial case |
| 47 | 103 | 32 | 37.4 | 21.36 | 17.48 | 61.17 | Not citizen science—Clinical trial |

outcomes of scientific activities is considered important, yet does not make it a citizen science activity: 'Staying up with the field is worthy, as is supporting its programs financially, but being a scientist entails doing science: observe, hypothesize, test, observe, predict, test, etc. Just as I support art without being an artist, Jane is supporting science' (Cliff Tyllick). Additionally, issues relating to ownership and power came to the fore in the comments. As one comment put it: '[the participant] does not have direct control or power over the science or the data—which disqualifies it for citizen science' (anonymous).

*Gaming environment & commercial leadership* (*Vignette 12*) presented the highest ambiguity and is the fifth most 'not easy' vignette to judge (that is, received either 'somewhat complex' or 'difficult' rating for complexity), due to two critical factors. Firstly, the lack of awareness of the participants that they are partaking in a scientific activity, raised many reservations towards this being citizen science. This form of unconscious participation is paired with both a lack of cognitive engagement and a lack of scientific interest, which, according to many comments, seems to be a requisite for citizen science. Secondly, the participant has to pay for the activity. Comments showed that participation fees have a negative connotation, which is sometimes linked to the activities embedded in a gaming format, such as 'being tricked to supply qualified work on the cheap' (anonymous), or associated with a company that has profit goals. The gaming approach itself is regarded both positively and negatively. For instance, some comments identified the activity as a well-known citizen science activity that makes use of a gaming approach, while one comment downgraded it because of this format: 'This is a game. A spinoff of the game is the science' (Tony Rebelo).

*Reuse of social media & public authorities data* (*Vignette 13*) has an average rating of below 50 and many ratings of 45. The comments mainly discuss two related areas of conflict. Firstly, the participant is not informed that her photos posted on social media are used within a scientific activity to identify invasive insect species. Therefore, she can neither be assumed to have participated voluntarily nor have been motivated by the research project, as she simply remains uninformed. 'Dorota does not intentionally contribute to science' (Sven Schade), and 'she has no opportunity to learn and interact' (Aleksandra Hebda). Secondly, this is accompanied by the critical assessment that the participant is not involved in the research process, despite the fact that her work (produced in a different context and with different intentions) has become a source of data. This lack of connection between the motivation for taking the photograph and its use in the research process highlights the issue. Conversely, the vignette highlights that citizen science implies conscious activity and that the participants are motivated and informed to participate in the research process. The use of passive data resources was ranked low across all vignettes.

*Commercial purpose, data and leadership* (*Vignette 15*) was viewed as being the most 'difficult' to judge (with 14.3% indicating this) and received a low percentage of ranking of the extremes (0 or 100), with an overall average close to the middle (54.24). The commercialization factor caused many to rate this vignette very low: 'if data were public and for scientific research and not commercial use, it would have scored 100%' (anonymous), with very few respondents comfortable with a company making profit out of this activity. Despite the participants collecting data—an activity that would have been described under other circumstances as being core to citizen science—the vignette was rejected as citizen science. One of the main reasons for this rejection was the concern that data and results are not freely available. For many respondents, this non-compliance with the open access expectation is a strong argument for deciding that it is not, or only to a limited degree, citizen science. Related to this is the argument that it is not a public matter but is motivated by business logic and lacks scientific rigour: often the research question is missing and participants are not involved in the entire research process. The lack of active participation and application of cognitive engagement were also given as reasons for the low rating. Respondents stressed the point that the measuring and evaluation of devices and computer systems is taken over by the company. Thus, it remains unclear how much knowledge and, therefore, how actively the participant is involved in the process.

*Clear citizen science* (*Vignette 41*) received a high average rating of 80.2. This vignette also received the fewest text comments explaining a decision (22% of respondents gave an explanation). The activities that the protagonist Femke carries out, namely a specific kind of data analysis as part of the classification of galaxies, was indeed called 'a classic citizen science project' (anonymous). Nevertheless, this vignette was not rated 100 and was criticized for the limited involvement of the participants—more specifically, the lack of integration of citizens in the other phases of the research process; the lack of insight and knowledge gained by the participating citizens; and the lack of sufficient acknowledgement of the work carried out by the participants in scientific publications. One comment stated: 'but if scientists would pay tribute to the highly qualified amount of workers, I would rate

them even higher than CS' (Katja Mayer). Some comments on this vignette also make a reference to artificial intelligence (AI): 'If this counts as citizen science, we should also treat Google CAPTCHAs as citizen science' (anonymous). This indicates that citizen science may be considered to play a role in the (further) development of AI.

*Non-controversial case—Galaxy Zoo (Vignette 45)* is recognized as the clearest example of citizen science with the highest number of 100 ratings. Two anonymous comments called the vignette 'textbook citizen science' (anonymous), others call it a 'classical example of citizen science' (Rosina Malagrida). Participants are involved in the different steps and phases of a research project: the protagonist collects data, analyses his data, undergoes further training and is part of scientific publications. The comments of those who rated this vignette lower criticize that the participant was not involved in the conception and initiation of the activity: 'He's participating in all stages other than the setting of goals and design of the experiment. If he did those things too, then I'd rate it all the way' (Jonathan Long). From these comments, it becomes clear that people tend to rate vignettes higher if the participants are involved in more steps of the research process. Two comments reflected this view in an interesting way: on the one hand, Tony Rebelo doubts that it is realistic to have participants equally involved in all phases; on the other hand, one comment interprets this vignette as a substantial expansion of the role of citizens: 'This story also shows elements of an as-yet-unnamed role: the citizen scientist who becomes attached to the project and participates not just in the citizen science role, but as a person who takes responsibility for improving/extending the project, e.g. by recruiting, mentoring, consulting, etc.' (anonymous).

*Not citizen science—clinical trial (Vignette 47)* is the most unusual vignette as it was designed to be clearly outside the scope of citizen science and was indeed ranked as 'not citizen science' overall, and yet, 17.5% of respondents still gave it a rating of 100. It is an example where comments are not only based on the text in the vignette but also on how such a case as presented in the vignette could be developed into a citizen science activity. We found comments that refer to the limited level of engagement, indicating that citizen science needs to include participants in various steps of the scientific process; some comments even provided suggestions of how to raise the level of engagement. While most comments identified the participant as an object of a medical study, we also found expressions of uncertainty about where to draw the line: 'Research on humans is always blurrier because there is a line between citizen scientist and research subject somewhere and it isn't always clear' (anonymous). The fact that an app is used to collect and report the data also influenced the low rating, indicating that the mere use of technology does not turn a project into citizen science: 'Is it just the case that we now have devices that we can carry around at home and in our spare time that defines if we are doing citizen science???' (anonymous). The main arguments for classifying this vignette as citizen science stressed the fact that the citizen collects and contributes data to a scientific activity as well as gets free access to the results of the study. In summary, the justifications for a lower rating of this vignette revolved around the motive that the citizen provides data but is not involved in the study beyond that, while the higher ratings referred to data collection as being a core citizen science practice.

## 5.3. Summary of qualitative analysis

The diversity of the various selected vignettes represents the multiple dimensions in which citizen science activities take place, and the qualitative answers reveal superordinate or traditional patterns of argumentation:

Firstly, there is the issue of conscious, active and motivated participation in a scientific process. Based on the survey findings, participants in citizen science must be informed and consciously choose to participate, and thus be aware that they are participating in scientific issues and practices. Some survey comments go beyond this and demand from participants an explicit intention and motivation to participate in science even if this is difficult to determine and evaluate. This means that the vignettes in which participants were either uninformed about what they were doing or could be assumed to have no explicit interest in contribution to science of their own were rated lower. Arguably, while the duty to inform lies with the activity initiators and project staff, the motivation relates to the participants.

Secondly, comments raised the importance of the engagement of participants in multiple phases of the scientific process. Vignettes in which the participants are involved in all or many phases of the project (conception, data collection, data recording, organizing data and evaluations, as well as publishing) received higher ratings in comparison to more ambiguous cases. This can be an indication

of a trend of moving towards more comprehensive integration of participants (citizen scientists) into the research process.

Thirdly, a central recurring theme was the reference to data collection as a core citizen science activity. If an activity includes data collection, it was rated higher on average, even if other facets (e.g. appreciation of citizens in publications or integration in data conception and initiation) appeared to be less significant in the activity as described in the vignette.

Finally, in some respondents' comments, educational aspects were mentioned and vignettes were ranked lower in which no clear learning was discernible on the side of the participants. Citizen science is associated with aspects of learning and the increase in knowledge can take different forms, e.g. taking part in webinars, self-education and peer-to-peer learning while participating in the activity or structured training by activity owners.

# 6. Conclusion: the pluralities of citizen science

The purpose of our survey was to identify, demonstrate and investigate the range of views of what can constitute citizen science, rather than reflecting individual opinions or biases. Across the vignettes that we analysed only a few were widely agreed as being 'citizen science' or 'not citizen science'. This indicates that context-specific definitions are needed and a set of characteristics reflecting the diversity of opinions about citizen science can be beneficial [5].

Our analysis shows that the understanding of citizen science varies significantly, as do the confidence levels of the respondents when making a decision regarding whether a particular vignette should be considered citizen science or not. In 24 of 50 vignettes, the respondents' decisions are linked with the varying levels of confidence. In these cases, confident respondents tended to assign vignettes more vigorously to either end of the spectrum, while less confident respondents verged towards the average rating. However, even for experienced respondents, making a decision on whether a vignette is citizen science or not was not easy, and this is expressed in the ranking that they provided, since ranking that is close to 50 indicates indecisiveness.

The comments provided by respondents for various vignettes also showed a wide array of interpretations as well as clear disagreements. For example, V12 describes a game that was created by a private company in collaboration with scientists, and in which participants become involved after paying a subscription fee. This case resulted in a high degree of ambiguity. It was the highest-ranking case considered to be 'not easy' to categorize as citizen science, with some comments arguing that a subscription fee meant the research was not citizen science. Others suggested that merely contributing to scientific processes was the key criterion for rating this citizen science.

In this paper, we have shared the results of a novel vignette study to understand the views and perspectives on what can be considered citizen science. Through a process of reviewing the literature for typologies and definitions, we identified a set of 10 factors and 61 sub-factors that can influence the judgement regarding a specific activity. These factors served as the basis for the development of 50 short vignettes which describe an activity. Of these 50 vignettes, five were designed to fit the consensus in the literature about what is citizen science, and another five selected according to statements that they should not be considered as citizen science. The vignette study used a crowdsourcing framework, allowing respondents to answer as many examples as they wished.

The 333 responses provided 5155 ratings for the 50 different vignettes, making use of the full range of possible degrees of citizen science. In addition, extensive textual information was provided by the respondents. The results demonstrate the plurality of views in relations to the literature and highlight some major issues that need to be considered in the design and implementation of citizen science activities, including the need to consider power relationships, the role of commercial actors, and the impact of payment from and to participants. The results of the survey provided the foundation for the compilation of the ECSA characteristics of citizen science [5] and were translated into a set of characteristics. We focused on the ambiguities in the field using examples and cases for areas where a wide range of opinions and disagreements exist (see §1). We then grouped these grey areas into five categories: (i) core concepts, (ii) disciplinary aspects, (iii) leadership and participation, (iv) financial aspects, and (v) data and knowledge. Through an inclusive process, both during the implementation of the survey and while producing the results, we created a comprehensive set of characteristics to ensure that different stakeholders could use our findings as the basis for their specific contexts and purposes. For instance, a funder that aims to finance an environmental citizen science initiative may have different requirements regarding data quality than a civil-society organization (CSO) aiming to

work with indigenous communities to include local knowledge in policy-making processes. The process of using the survey for the development of the characteristics is beyond the scope of this paper.

Along with the quantitative results, we also addressed the additional comments that the respondents provided in the characteristics, as each case produced a significant number of qualitative outcomes that were quite diverse, bringing both clarity and richness to our study. Additionally, the ECSA 10 Principles of Citizen Science have been an important guideline for how we laid out the results of the survey in the characteristics, because they provide more concrete examples for these very broad and comprehensive 10 principles [6]. Therefore, we recommend that the characteristics should be considered together with the ECSA 10 Principles of Citizen Science.

Our analysis looked at the results in aggregate and there is scope for further studies of this dataset in identifying clustering of opinions and positions, as well as identifying disciplinary perspectives or regional variations. The survey was carried out only in English, and future research can include the translation of the vignettes to several languages, in order to understand how different language communities perceive the field.

Overall, this study shows that there is a need to further address the plurality and diversity of interpretations in the field of citizen science. This includes not only a definition of citizen science, but also the diverse typologies and terms that are used to describe citizen science based on the context, scientific discipline and geography from which they originate. This clearly indicates that context-specificity, openness and fluidity of definitions reflect the diversity of the field and leave the necessary space for methodological advancements, disciplinary cross-fertilization and overall, for the growth of citizen science as a field for science innovation.

Ethics. The survey followed data minimization practices and the analysis did not include personal details, except where the respondents explicitly asked to be associated with their contribution. Data protection registration for the project was provided by the Centre for Research and Interdisciplinarity, Université de Paris. All survey respondents who are mentioned by name have explicitly provided their consent to be associated with the comments and to retain their name in the deposited file.

Data accessibility. The vignettes are available on Zenodo at https://doi.org/10.5281/zenodo.4281293. The survey results are accessible on Zenodo under https://doi.org/10.5281/zenodo.4266685

Authors' contributions. The study was initiated by M.H., M.G. and Ma.Ma.; with support from D.F., S.H., L.C., B.K., S.R. and D.R. The design workshops were organized by M.H., M.G., G.H., S.H., U.W. and Ma.Ma.; with support from L.C., B.K., C.N., K.W., A.M., D.R., T.S. and M.W. The framework was designed by M.H., D.F., M.G., G.H., S.H., B.K. and C.N.; with support from M.G., L.C., K.W. and Mo.Ma. The vignettes were written by M.H., D.F., M.G., G.H., S.H., B.K. and C.N.; with support from B.G.T., L.C. and K.W. The survey implementation was done by M.H.; with support from B.G.T. The survey promotion and outreach were done by M.H., M.G. and M.W.; with support from B.G.T., S.H., U.W., B.B., A.S., D.R., D.D., F.H. and A.L. The survey analysis was done by M.H., D.F., B.G.T. and S.H.; with support from M.G., G.H., L.C., B.K., S.W. and T.S. The characteristics were written by M.H., D.F., G.H., L.C., B.K., S.W., Ma.Ma., L.A.S., D.R. and T.S.; with support from B.G.T., M.G., S.H., U.W., C.N., S.R. and K.W. The manuscript was structured by M.H., D.F., B.G.T. and G.H.; with support from S.H. The lead editing of different sections was done by: M.H., D.F., B.G.T., M.G., S.H., L.C., S.W. and A.M. The *introduction and background* sections were written by M.H., M.G., G.H., L.C., S.W., L.A.S. and A.M.; with support from B.G.T., U.W., C.N., A.S., D.D., F.H. and A.L. The *methodology* section was written by M.H. and G.H.; with support from S.H. and U.W. The *survey respondent characteristics* section was written by M.H. and B.G.T.; with support from D.F. and S.R. The *views about citizen science* section was written by B.G.T.; with support from M.H., D.F., L.C. and S.R. The *content analysis of textual responses* section was written by S.H., B.K. and K.W.; with support from M.H., D.F., B.G.T. and B.B. The *conclusions* section was written by D.F., M.G. and C.N.; with support from M.H., B.G.T., G.H. and A.S. The final editing was done by M.H., D.F., B.G.T., M.G., G.H., S.H., B.K., U.W., S.W., B.B., S.F., L.A.S. and A.S.; with support by A.L.

Competing interests. We declare we have no competing interests.

Funding. The development of these characteristics was supported by European Union's Horizon 2020 research and innovation programme under grant agreement no. 824580, project EU-Citizen.Science (The Platform for Sharing, Initiating, and Learning Citizen Science in Europe), the ERC Advanced Grant project European Citizen Science: Analysis and Visualisation (under grant agreement no. 694767). Thanks to the Bettencourt Schueller Foundation long-term partnership, this work was partly supported by CRI Research Fellowships to Muki Haklay, Alice Motion and Bastian Greshake Tzovaras.

Acknowledgements. We would like to thank Lionel Deveaux for setting up the survey and Soledad Luna, Fredrik Brounéus, Katja Heuer and Tim Woods for promoting the survey, providing feedback on earlier versions of the vignettes and assisting with the development of this paper. We would like to thank the following people who contributed to the survey: Hilal Us, Simon, Sarah Angulo, Didone Frigerio, Alex Richter-Boix, Anna-Lena Maedge, Bernhard Arnold, Juan Manuel Rosso, Sigrid Peter, Sven Schade, Carolina, Renee Sieber, Femke Tomas, Gabriel Dean, Jonathan Long Rosina Malagrida, Joao Cao Duarte,Inger Elisabeth Maaren, Larissa Braz Sousa, Diana Marques, Ramona j Wysong, Gina Agostini, Mario Pesch, Kristin Oswald, Ana I. Faustino, Jelena Hessel, Anna

Krzywoszynska, Maria Bostenaru Dan, Anna Berti Suman, Daphne Hoh, Michael Poulsen, Aleksandra Hebda, Sarah de Launey, Laura Seelen, Katja Mayer, Tim Adriaens, Samboko Winter, Eva Lewandowski, Giovanna Flaim, Stefania Pinna, Elisabetta Broglio, Victoria J. Burton, Nicolas M. Adreani, Francois Bry, Liesbeth Gijsel, Anne-Sophie, Ravaud, Tony Rebelo, Anirudh Krishnakumar, Helen Whitehead, Dr Jane A. Chukwudebelu, Cliff Tyllick, Karolina Alexiou, Yaqian Wu, C. McLeod, Claire Ellul, Katrin Bohn, Tahani Nadim, Sara Riggare, Uwe Moldrzyk, Karin Ekman, Giulia Melilli, Pamela Buchan, Sarah Speed, Hannah Lacey, Stijn Calders, Simon Etter, Vanessa van den Bogaert, Julian Vicens and Amber Griffiths.

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
