## [Peer Review File · Royal Society Open Science]

Review History

RSOS-202108.R0 (Original submission)

Review form: Reviewer 1

Is the manuscript scientifically sound in its present form?

Yes

Are the interpretations and conclusions justified by the results?

Yes

Is the language acceptable?

Yes

Do you have any ethical concerns with this paper?

No

Have you any concerns about statistical analyses in this paper?

No

Recommendation?

Accept with minor revision (please list in comments)

Comments to the Author(s)

It was an absolute pleasure to read this manuscript "Contours of citizen science". One of the most well-executed papers I've read in some time, and will definitely help the field of citizen science and also contribute to our understanding of what that "field" looks like! I think the figures, interpretation, and conclusions were all excellent. I also like how they treated the term 'citizen science' in the introduction. A great perspective.

Only because I feel like I should have at least one "comment"...

When I was reading through the introduction, it didn't jump out at me who the audience was of the survey. It looks like it was somewhat general, but in the end it was mostly people like myself I assume who work in the field of citizen science - either running projects, analyzing data, academics etc. And this is a source of bias in the survey participants results. This is totally fine, but I think just potentially worth highlighting how this was set up and why these were the targets. On a side note, would be interesting to repeat this, but target the contributors to the citizen science projects a bit more heavily!

Regardless, great stuff, and I look forward to citing it in my future work.

Review form: Reviewer 2

Is the manuscript scientifically sound in its present form?

Yes

Are the interpretations and conclusions justified by the results?

Yes

Is the language acceptable?

Yes

Do you have any ethical concerns with this paper?

No

Have you any concerns about statistical analyses in this paper?

No

Recommendation?

Accept with minor revision (please list in comments)

Comments to the Author(s)

1. The justification for the use of vignette is not convincing enough because the paper failed to present other available options.
2. The use of past experience without mention of personal characteristics as basis of judgment need to be explained.

Decision letter (RSOS-202108.R0)

Dear Dr Haklay

On behalf of the Editors, we are pleased to inform you that your Manuscript RSOS-202108 "Contours of citizen science: a vignette study" has been accepted for publication in Royal Society Open Science subject to minor revision in accordance with the referees' reports. Please find the referees' comments along with any feedback from the Editors below my signature. Please accept our apologies the lengthy review process.

Please submit your revised manuscript and required files (see below) no later than 7 days from today's (ie 14-Jul-2021) date. Note: the ScholarOne system will 'lock' if submission of the revision is attempted 7 or more days after the deadline. If you do not think you will be able to meet this deadline please contact the editorial office immediately.

on behalf of Prof Pete Smith (Subject Editor)
openscience@royalsociety.org

Reviewer comments to Author:

Reviewer: 1

Comments to the Author(s)

It was an absolute pleasure to read this manuscript "Contours of citizen science". One of the most well-executed papers I've read in some time, and will definitely help the field of citizen science and also contribute to our understanding of what that "field" looks like! I think the figures, interpretation, and conclusions were all excellent. I also like how they treated the term 'citizen science' in the introduction. A great perspective.

Only because I feel like I should have at least one "comment" ...

When I was reading through the introduction, it didn't jump out at me who the audience was of the survey. It looks like it was somewhat general, but in the end it was mostly people like myself I assume who work in the field of citizen science - either running projects, analyzing data, academics etc. And this is a source of bias in the survey participants results. This is totally fine, but I think just potentially worth highlighting how this was set up and why these were the targets. On a side note, would be interesting to repeat this, but target the contributors to the citizen science projects a bit more heavily!

Regardless, great stuff, and I look forward to citing it in my future work.

Reviewer: 2

Comments to the Author(s)

1. The justification for the use of vignette is not convincing enough because the paper failed to present other available options.
2. The use of past experience without mention of personal characteristics as basis of judgment need to be explained.

===PREPARING YOUR MANUSCRIPT===

===PREPARING YOUR REVISION IN SCHOLARONE===

To revise your manuscript, log into <https://mc.manuscriptcentral.com/rsos> and enter your Author Centre - this may be accessed by clicking on "Author" in the dark toolbar at the top of the

page (just below the journal name). You will find your manuscript listed under "Manuscripts with Decisions". Under "Actions", click on "Create a Revision".

<https://royalsociety.org/journals/authors/author-guidelines/#supplementary-material> to include a suitable title and informative caption. An example of appropriate titling and captioning may be found at https://figshare.com/articles/Table_S2_from_Is_there_a_trade-off_between_peak_performance_and_performance_breadth_across_temperatures_for_aerobic_sc_ope_in_teleost_fishes_/3843624.

Author's Response to Decision Letter for (RSOS-202108.R0)

See Appendix A.

Decision letter (RSOS-202108.R1)

Dear Dr Haklay,

I am pleased to inform you that your manuscript entitled "Contours of citizen science: a vignette study" is now accepted for publication in Royal Society Open Science.

on behalf of Pete Smith (Subject Editor)
openscience@royalsociety.org

Appendix A

Dear Editorial Team of Royal Society Open Science,

Many thanks for your handling of the reviewing process of this paper. We are pleased to read that the reviewers have appreciated the paper and provided useful comments that are helping us with improving the final version of the paper.

We have provided the following files. The file `author_tex.tex` is an editable version of the paper. The changes that were made to the paper can be found searching “Revised text” in the document. To assist this process, we provide two PDFs. In them, you’ll find that we have addressed Reviewer 1 comments on Pages 2 and 13 (bottom of the page), while Reviewer 2 comments are addressed on pages 10 (justification for the methodology) and 13.

We uploaded all the images as EPS files and that are associated with the TeX file.

The tables are uploaded as Word document. However, notice that for Table 1 and Table 3, we have used the BibTeX format for referencing the source. As you will see in the PDF, this is handled appropriately in the TeX compilation.

All the files that are needed for the production are also available at <https://www.overleaf.com/7476765929pnvqdvrszsn> and we will be happy to assist you in any way to see this paper published.

Many thanks for your work!

Muki Haklay and the authors.